# CMT-SCTP and MPTCP Multipath Transport Protocols: A Comprehensive Review

**Parul Tomar [1], Gyanendra Kumar [1,2], Lal Pratap Verma [3], Varun Kumar Sharma [4], Dimitris Kanellopoulos [5,\*], Sur Singh Rawat [6] and Youseef Alotaibi [7]**

1 Department of Computer Engineering, J.C. Bose University of Science and Technology, YMCA, Faridabad 121006, India; ptomar_p@hotmail.com (P.T.); gyanendrakumar@galgotiasuniversity.edu.in (G.K.)
2 Department of Computer Science and Engineering, Galgotias University, Greater Noida 201310, India
3 Department of Computer and Communication Engineering, Manipal University Jaipur, Jaipur 303007, India; er.lpverma1986@gmail.com
4 Department of Computer Science and Engineering, The LNM Institute of Information Technology, Jaipur 302031, India; varunksharma.102119.cse@gmail.com
5 Department of Mathematics, University of Patras, 26500 Patras GR, Greece
6 Department of Computer Science and Engineering, J.S.S. Academy of Technical Education, Noida 201301, India; sur.rawat@jssaten.ac.in
7 Department of Computer Science, College of Computer and Information Systems, Umm Al-Qura University, Makkah 21955, Saudi Arabia; yaotaibi@uqu.edu.sa
* Correspondence: d_kan2006@yahoo.gr

**Abstract:** A huge amount of generated data is regularly exploding into the network by the users through smartphones, laptops, tablets, self-configured Internet-of-things (IoT) devices, and machine-to-machine (M2M) communication. In such a situation, satisfying critical quality-of-service (QoS) requirements (e.g., throughput, latency, bandwidth, and reliability) is a large challenge as a vast amount of data travels into the network. Nowadays, strict QoS requirements must be satisfied efficiently in many networked multimedia applications when intelligent multi-homed devices are used. Such devices support the concept of multi-homing. To be precise, they have multiple network interfaces that aim to connect and communicate concurrently with different networking technologies. Therefore, many multipath transport protocols are provided to multi-homed devices, which aim (1) to take advantage of several network paths at the transport layer (Layer-4) and (2) to meet the strict QoS requirements for providing low network latency, higher data rates, and increased reliability. To this end, this survey first presents the challenges/problems for supporting multipath transmission with possible solutions. Then, it reviews recent research efforts related to the concurrent multipath transmission (CMT) protocol and the multipath transmission control protocol (MPTCP). It reviews the latest research efforts by considering (1) how a multipath transport protocol operates (i.e., its functionality); (2) in what type of network; (3) what path characteristics it should consider; and (4) how it addresses various design challenges. Furthermore, it presents some lessons learned and discusses open research issues in multipath transport protocols.

**Keywords:** CMT-SCTP; MPTCP; IoT; multi-homed user devices; multipath communication

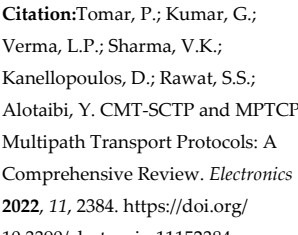

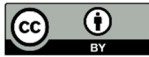

## 1. Introduction

Nowadays, the support of high-quality video streaming services in a wireless mobile network is very demanding, as throughput can be degraded along with the traffic load, attenuation loss, fading, and signal-to-noise ratio. Meanwhile, current mobile devices have a large storage size, high computing power, high-resolution display capability, and multiple sophisticated networking interfaces. It is expected that these mobile devices will improve the QoS by simultaneously using multiple networking

interfaces. A multi-homed user device such as a smartphone, laptop, tablet, or IoT device can support multiple networking interfaces such as Wi-Fi (IEEE 802.11), cellular interface (3G/4G/LTE), and Ethernet. As shown in Figure 1, it can be connected simultaneously using numerous network access technologies through different pathways (disjoint paths).

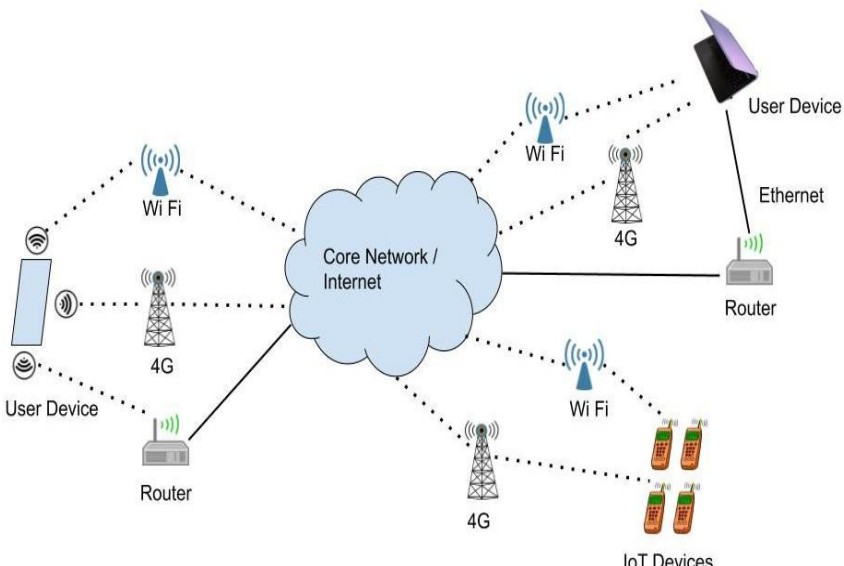

**Figure 1.** Multi-homed user devices.

The multi-homing feature [1–3] can improve the network's reliability, resilience, load balancing, and fault tolerance. Multi-homing is a cost-effective, technically feasible, and widely accepted capability of user devices. The plethora of multi-homed devices and the advances of the fifth-generation (5G) cellular networks have efficiently utilized the available resources of multi-communication interfaces. The diversity of M2M and IoT-based applications mainly operate in resource-constrained network environments such as radio links, Wi-Fi, and 4G/Long-Term-Evolution (LTE) networks. In such network environments, multipath transport protocols exploit multiple available paths and struggle to fulfill the strict QoS constraints for higher data rates, low network latency, and high reliability. A solution is the optimal use of the available multiple networking interfaces of multi-homed user devices.

The transmission control protocol (TCP) does not provide multi-homing. In particular, an application can only attach a single IP address to one specific TCP connection with another host. If the TCP sockets-based application programming interface (API) connected with that IP address breaks down, the TCP connection must be reestablished as it is missing. For this reason, the internet engineering task force (IETF) standardized the stream control transmission protocol (SCTP) [4] to integrate the multi-homing feature into its specification. Then, Iyengar et al. [5] designed the CMT approach for multi-network interface devices to utilize such a feature of SCTP. CMT (stated as CMT-SCTP) is based on SCTP and improves throughput, resource utilization, latency, and network reliability. However, CMT-SCTP suffers from severe drawbacks: inappropriate packet scheduling, needless packet retransmission, unnecessary reduction of the *congestion window* (CWND), receiver buffer blocking (RBB) [6,7], and so on.

The multipath TCP working group of IETF introduced the multipath TCP (MPTCP) [8,9], which allows a TCP connection to employ multiple paths to exploit resource usage and enhance redundancy. In MPTCP, a multipath connection that contains multiple flows can dynamically be established. As depicted in Figure 2a, MPTCP transfers data simultaneously over different accessible subflows, including IPv4 and IPv6. As shown in

Figure 2b, the MPTCP protocol has two main functions: (1) path management (PM), and (2) multipath packet scheduling (PS).

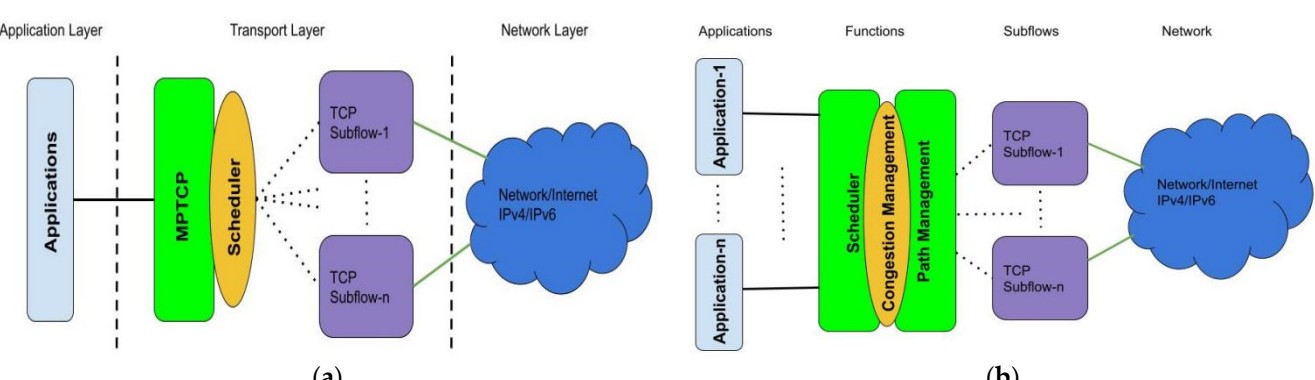

**Figure 2.** The architecture of MPTCP. (**a**) Protocol stack. (**b**) Functions.

The primary mission of PM is to establish, remove, and control those subflows that can play a part in the end-to-end transmission of data packets. The PM algorithm can dynamically add or delete subflows to participate in the concurrent transmission. PM initiates and manages the subflows, which are part of the same multipath connection. On the other hand, a multipath packet scheduler distributes packets over different paths according to a particular policy. For example, a packet scheduler can aggregate throughput by utilizing all available capacity. Moreover, a packet scheduler can reduce latency by choosing low latency paths or can enhance reliability (duplicate packets). Apart from the multipath scheduler, MPTCP has an additional flow management mechanism, *congestion control* (CC), that controls induced network load and subsequently avoids congestion.

In a wireless context, the performance of MPTCP is mainly limited due to the long round-trip-time (RTT) and the frequent loss of data packets. For this reason, various multipath transport protocols have been suggested in the past. CMT-SCTP [4,5] and MPTCP [8,9] protocols constitute a boon for multipath data transmission devices. Multipathing solves the problem of single-path insufficiency by combining multiple available pathways to increase bandwidth and throughput [10–12].

## 2. Scope and Contribution of This Survey

Xu et al. [13] surveyed various CC algorithms for multipath transport protocols. Further, they discussed how a multipath CC scheme must be designed to address the need for some desirable properties such as load balancing, TCP-friendliness, stability, and Pareto optimality. Notably, these properties are discussed later in Section 4. The authors [13] investigated existing window-based and rate-based multipath CC algorithms. Siddiqi et al. [14] investigated the latest research contributions on learning-based CC algorithms that control data traffic in MPTCP. In particular, the authors focused on deep reinforcement learning (DRL)-based CC algorithms.

MPTCP has been used in a limited domain due to its low adaptability to heterogeneous networks. The real reason is that it is a difficult task to design, implement, and test MPTCP on real mobile smart devices (MSDs). Zhang et al. [15] methodically studied MPTCP and clearly described the relationship between each portion of MPTCP. Furthermore, they proposed an original solution to port MPTCP to MSDs. From another perspective, Jagetiya et al. [16] observed the suitability of MPTCP in scenarios where multi-homed devices use homogeneous or heterogeneous network interfaces, taking for granted that an MPTCP-enabled client is connected to an MPTCP-enabled server through two network interfaces. Chao et al. [17] surveyed offered MPTCP works and presented a

summary of multipath routing. This is the first work that presents the recent progress of MPTCP in vehicular ad hoc networks (VANETs) and multipath routing in VANETs.

Recently, a single-user terminal can be connected to multiple radio access points due to the *multi-connectivity* of the 5G cellular networks. 5G multi-connectivity [18] supports simultaneous connectivity and aggregation across different types of technologies such as 5G and 4G, as well as unlicensed technologies such as IEEE 802.11 (Wi-Fi). To enable 5G multi-connectivity, the 3rd Generation Partnership Project (3GPP) recently proposed the access traffic steering, switching, and splitting (ATSSS) architecture [19]. In the ATSSS architecture, a key technology enabler is multipath transport protocols. To this end, Wu et al. [20] reviewed the 5G background and in-progress standardization activities around multi-connectivity and the ATSSS architecture. They also reviewed multipath transport protocols for 5G, subjected to the standardized ATSSS architecture.

This survey paper focuses on connection-oriented multipath protocols located at Layer-4 of the Open Systems Interconnection-Reference Model (OSI-RM). We categorize them by considering their salient features along with their applicability. Notably, we do not present the variants of multipath QUIC (MQUIC), even though MQUIC [21] is an alternative multipath transport protocol in the ATSSS architecture. The paper's contributions are as follows:

1. It presents the main challenges/problems that arise in multipath transmission and their suggested solutions.
2. It presents a comprehensive study of the existing CMT-SCTP and MPTCP multipath transport protocols.
3. It qualitatively compares and evaluates CMT-SCTP and MPTCP transport protocols.
4. It highlights future research directions for CMT-SCTP and MPTCP transport protocols.

The paper is structured as follows. Section 3 discusses the main applications and advantages of multipath transport protocols. Section 4 presents the problems and technical challenges of supporting multipath communication. Section 5 analyses CMT-SCTP multipath protocols, while Section 6 analyses MPTCP multipath transport protocols. Both sections qualitatively compare the multipath protocols in different application scenarios and implementation contexts. Section 7 presents some lessons learned. Section 8 provides future research directions. Finally, Section 9 concludes the paper.

## 3. Multipath Transport Protocols: Main Applications and Advantages

Hereafter, the main applications and advantages of multipath transport protocols are discussed.

### 3.1. Main Applications

− *Selecting the most effective service plans*: In mobile devices, exploiting multiple pathways improves network stability and fault tolerance and allows users to take advantage of more cost-effective service plans. For example, users can choose the best and cheapest plan when 3G and Wi-Fi interfaces are available.

− *Data center networks*: Another critical application of a multipath transport protocol is the data center network. An enhanced multipath transport protocol allows a variety of network architectures in the data center that single-path Layer-4 protocols could not provide. A data center network architecture supports many network services. It usually offers thick interconnectivity in the network by managing multiple paths among servers and excessive aggregate bandwidth. For example, by leveraging the available bandwidth of several paths, Amazon Elastic Compute Cloud (EC2) achieves three times the performance of a single path [22,23].

*3.2. Main Advantages*

Hereafter, we present the significant advantages of multipath Layer-4 protocols.

− *Load balancing*: Load balancing can be carried out at the network layer. A network protocol can route data packets on the basis of different path load states. However, a load balancing technique at the network layer may cause network instability [24]. On the other hand, a load balancing technique at Layer-4 progressively raises traffic rates on each available path (across multiple RTTs). Further, such a technique steadily balances traffic (at each way).

− *Resource pooling*: At Layer-4, resource pooling of many pathways allows for greater exploitation of path characteristics (bandwidth, latency, and RTT) than a single link [25]. Layer-4 resource pooling enables the network to flexibly distribute available resources to meet current traffic demands. The application traffic gets disrupted when a primary path breaks in single-path applications. On the other hand, multipath routing redirects traffic from problematic pathways to other available paths.

− *Path diversity*: Network diversity is a resource utilization strategy used in data center networks, wireless networks, and the Internet. According to [26], an alternate path with superior bandwidth and transmission latency is available compared to the default path in 30–80% of circumstances. As a result, reliability and bandwidth aggregation may be accomplished by utilizing multipath diversity. In multimedia applications, path diversity is beneficial for decreasing packet loss [27] and end-to-end latency [28].

− *Key role in future Internet technologies*: Multipath at Layer-4 allows users to switch from one access technology to another (3G to Wi-Fi) [29]. Multipath is expected to play a key role in the evolution of future Internet technologies and cloud computing. Aside from that, the objective for 5G technology is to leverage concurrent multipath data transmission technology to meet high-bandwidth and low-delay requirements. Moreover, multiple distinct channels offer improved throughput to satisfy the demand for cloud computing.

− *Increased security*: In multipath transmission, data packets get transmitted via multiple independent subflows. Consequently, it becomes difficult for intruders or malicious entities to intercept or monitor the data [30].

− *High throughput*: The network's throughput is essential in many real-time applications [31,32]. A high data transmission rate can be achieved by exploiting multiple network paths.

## 4. Challenges for Supporting Multipath Communication

Theoretically, multipath communication can increase network performance by utilizing available network resources. However, multipath communication faces the following technical challenges in practice: (1) multipath scheduling; (2) excessive network congestion; (3) CWND growth policy for CC; (4) packet loss and retransmission; (5) the RBB; (6) packet reordering;(7) the head of line (HoL) blocking; (8) stream handling; (9) long RTT; (10) channel impairment; (11) heterogeneous communication standards; (12) the Pareto-optimality issue; and (13) various security issues. Hereafter, we analyze these challenges.

*Multipath scheduling***:** Concurrent data transmission over multiple interfaces does not necessarily meet the expectations. It happens because different paths with varying properties (e.g., the heterogeneous nature of transmission delay and bandwidth) cause the delivery of out-of-order data packets. Thus, they reduce the required performance of the network. In multipath transmission, intelligent scheduling of data packets over multiple available interfaces is necessary to improve the overall network performance. It has been observed that an efficient packet scheduling policy has a significant impact on system performance [33]. Many multipath transmission methods are being developed to maximize the network's optimality by using different scheduling. For example, an MPTCP packet scheduler, coupled with a CC algorithm, can schedule payload data onto

the existing subflows. MPTCP, like TCP, uses two state variables to keep the transmission between the sender and receiver. The first is the CWND, and the second is the *slow start threshold* (ssthresh). The sender starts sending data with the initial CWND value, and CWND increases exponentially until it reaches a threshold (ssthresh). Figure 3 shows an MPTCP packet scheduler coupled with CC. The CC algorithm chooses a CWND $w_n$ for path $n$, which controls the transmission rate of this path. Some paths are considered unavailable because they have their CWND full. Available are those paths whose CWNDs have room. The packet scheduler selects one of the available paths on the basis of the scheduling criteria, which have a significant result on the performance of MPTCP.

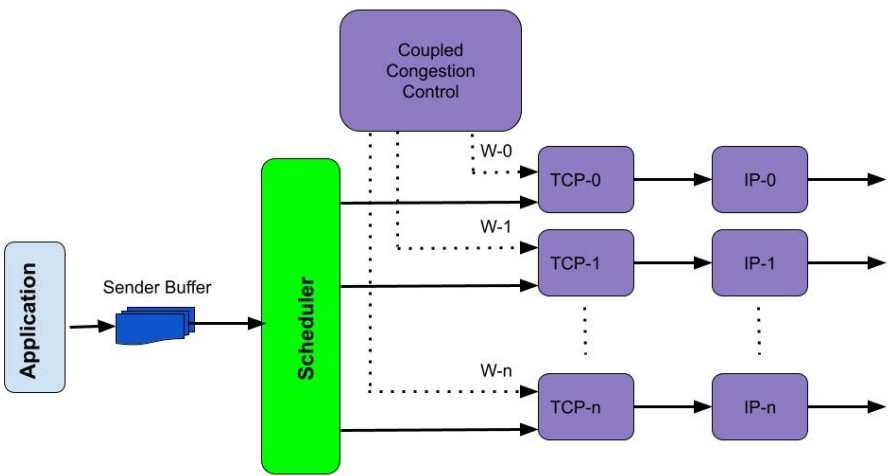

**Figure 3.** MPTCP packet scheduler coupled with CC.

Multipath schedulers [34] meet the demand for desirable properties such as load balancing, delay, bandwidth, QoS metrics, priority, and energy consumption. Consequently, many different types of multipath schedulers have been suggested. The scheduler can use RTT and CWND information of the OSI Layer-4 to estimate the transfer time of each packet on every path. On the other hand, the scheduler can provide low latency or can duplicate packets to offer high reliability. This need is determined by the optimization objective (latency or throughput) and the current path status. Moreover, machine learning methods, such as supervised or reinforcement learning (RL), can also be used to allow throughput and latency optimization in the same scheduling algorithm.

*Excessive network congestion:* One of the inefficiencies of multipath communication is excess data transmission due to multiple subflow management [35], increased acknowledgments (ACKs) [7], and frequent retransmission [36,37]. This inefficiency mandates multipath protocols to double ensure before generating data packets into the network to avoid excessive congestion.

*CWND growth policy for CC:* An efficient multipath scheduler is often coupled with a CC algorithm. The main objective of CWND is to limit the sender from sending more packets than the network capacity under the current load condition. The aim of changing the CWND is to adapt to the existing network status. Various authors suggested different CWND growth policies to adapt to network congestion status. For example, Iyengar et al. [5] suggested a solution for the CWND update problem, while Zhu et al. [38] presented a new CWND growth policy that uses a bandwidth estimation approach based on the link increases algorithm (BELIA) to enhance the throughput of multipath communication.

*Loss and retransmission:* When a multi-homed device transmits data packets through multiple paths, there is a high possibility of them being lost on the network. There can be many reasons for a packet being dropped in the network, such as a network error, a path disconnection, or a channel error. Iyengar et al. [5] rectified the retransmission issue in CMT and proposed the split fast retransmit (SFR) scheme to improve the network performance. From another perspective, Xu et al. [39] studied

networking problems and presented a new design of a deep RL-based control framework that realizes network experience to control congestion on MPTCP. Xu et al. [40] suggested a path quality-aware concurrent data transfer approach (CMT-QA) for transmitting data packets over a multipath and minimizing their retransmission. Cui et al. [41] identified the loss problem and path delay. They suggested the fountain code-based multipath TCP (FMTCP), which minimizes the effect of different path delays and losses. However, Peng et al. [42] proposed the *fluid* model for a large group of MPTCP schemes. This model improves TCP friendliness, responsiveness, and CWND growth.

*RBB*: In multipath communication, a sender transmits data concurrently over multiple paths to exploit the available network resources. However, each path has different transmission latency and capacity. Therefore, the receiver receives out-of-order data. The destination has a receiver buffer of limited size to reorder the received data. As the frequency of unordered data increases, the receiver buffer becomes blocked. Different techniques have been suggested to alleviate this problem. It is worth mentioning that CMT uses the equal data distribution policy to transfer data packets over the multiple available paths. Therefore, it creates an RBB problem due to dissimilar transmission latency and bandwidth of different paths. CMT suffers from the RBB problem, while every path has a dissimilar end-to-end delay and transmission rate. Yilmaz et al. [43] proposed a non-renegable selective ACK (NR-SACKs) to a free receiver-side buffer. This ACK strategy simply deletes the segment from the receiver buffer without considering the CWND growth and packet reordering. Natarajan et al. [44] suggested a new state, named *potentially failed*(PF), to mitigate the RBB problem that occurs due to path failure. This state shows whether the destination is reachable or not. Thus, all available new data packets will be forwarded to another alternative path. However, in [45], Xu et al. suggested the network coding-based CMT (CMT-NC) for minimizing the RBB problem in CMT-SCTP in the heterogeneous network environment.

*Packet reordering*: Reordering at the receiver end is one of the major issues in communication. It rarely occurs in single path transmission but frequently occurs in multipath transmission. Reordering in multipath transmission occurs due to packet loss and different delays of the different paths. Many solutions exist to handle it. In [41], the authors discussed four packet reordering solutions for MPTCP protocols. A packet reordering solution for the wireless environment is discussed in [46]. In [43,47], the authors presented reordering solutions for SCTP-based protocols.

*The HoL blocking problem*: To ensure ordered delivery of data packets, the receiver holds packets in the buffer until the lost packets are received. This condition is called *HoL blocking*[48–51]. Due to multiple subflows in multipath communication, the HoL blocking problem worsens. Therefore, the network performance is decreased. In [52–54], a forward error correction (FEC) coding scheme is proposed to handle the HoL blocking problem to recover from lost data packets at the receiver end. Another way to avoid HoL blocking is sending lost data packets via a faster subflow.

*Stream handling*: Firewalls, network address translators (NATs), and proxies are examples of network middleboxes. In a multipath transport protocol, a single data stream is transmitted via multiple subflows or network interfaces (e.g., Ethernet, Wi-Fi, and 5G/LTE). Notably, intermediary devices (i.e., network middleboxes) consider each subflow as a single TCP connection. Such consideration may force, for example, a gateway to change the data stream by adding or removing bytes to payloads. This results in changes in the boundaries of the data stream [55]. The multipath transport protocol should have provisions to detect these changes and fallback. A solution to this problem is an additional checksum on a payload with every segment of multipath communication [55,56]. This checksum allows for detecting any possible modification to the data stream.

*Long RTT*: Variations in RTT of the different paths may increase the arrival of unordered data packets at the receiver. Hence, the sender injects more duplicate data packets. More duplicate packets in the network system increase congestion and

contribute to longer RTT, leading to the system's poor performance. While transmitting data packets, a longer RTT of a path must be bounded to mitigate the arrival of expired and unordered packets.

*Communication channel impairment***:** Most communication interfaces in multi-homed devices are wireless and prone to error from external interferences. Examples of such interfaces are IEEE 802.11 (Wi-Fi), IEEE 802.15.1 (Bluetooth), Zigbee, and LTE [57]. The presence of external interference, interruption, path loss, and multipath fading may force the device to delay the transmission due to the false detection of the busy interface, packet loss, and jitter. Consequently, these factors degrade the performance of the multipath protocol.

*Heterogeneous communication standards***:** Standards for heterogeneous networks vary according to the adopted strategies of network carriers. For instance, the retransmission policy and packet loss handling in the 3G network are different from those in 4G [58]. Moreover, when 4G is compared to Wi-Fi, there is a difference in loss rate, RTT, and bandwidth due to different standards adopted by network carriers. Designing generic multipath protocols in such an environment is difficult without modifying existing standards. Many organizations, such as IEEE and IETF, are working to provide common security, privacy, communication, and architecture standards. However, more efforts are needed.

*Pareto-optimality***:** It is a key concept in the field of optimization. Considering resource pooling of a multipath transport protocol, Pareto-optimality is related to a situation in which an end-to-end connection increases its throughput without reducing the throughput of other coexisting end-to-end connections [59–61]. In this case, the multipath transport protocol is characterized as Pareto-optimal. It is a challenging task to design a multipath transport protocol as Pareto-optimal. Most multipath transport protocols are not Pareto optimal, such as the multipath protocol suggested in [62].

*Security issues***:** Various security threats have arisen due to the advent of the multipath communication paradigm. From the threats identified in [63,64], a multipath communication mechanism should provide secure handshaking, secure addition, and removal of multiple subflows, prevention from flooding, and hijacking attacks. An additional mechanism should be placed to tackle denial-of-service (DoS) attacks such as reset attacks and misuse of synchronization (SYN) packets/cookies. Applications also suffer from multiple IPs during multipath communication. A guideline is given in [65,66] to deal with such a problem. Another security breach that may arise from traffic splitting is the broken trust model. For example, analyzing intrusion detection and data leak prevention by the sniffer, firewall, or gateway may become problematic in multipath transmission [67].

Multipath connection-oriented protocols located at Layer-4 are divided into two categories:

(1) CMT-SCTP protocols which are based on SCTP; and
(2) MPTCP protocols which are based on TCP.

Figure 4 depicts CMT-SCTP and MPTCP variants related to their features.

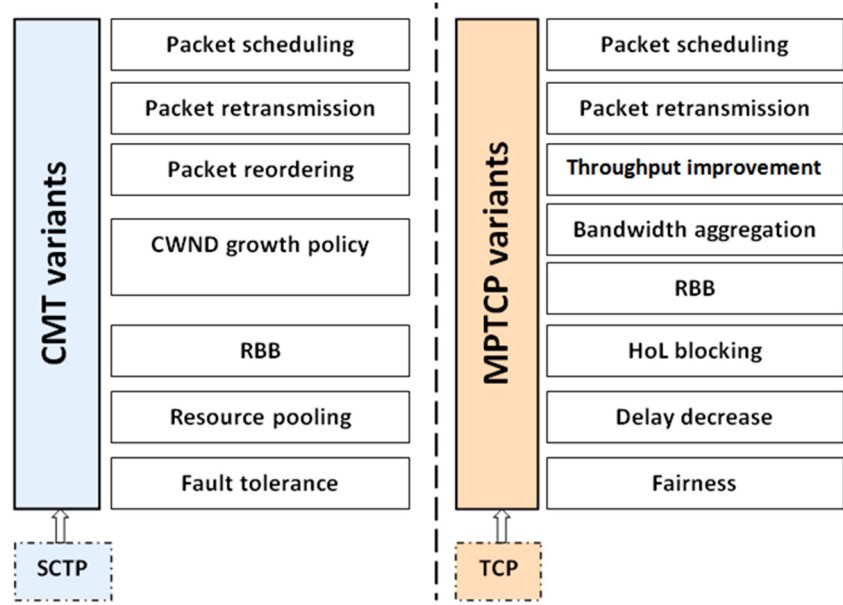

**Figure 4.** Multipath transport protocols related to their features.

Sections 5 and 6 analyze multipath transport protocols of both categories. These sections also present new multipath techniques/solutions (mainly suggested by IETF) that implement multi-homing and address problems in multipath communication. Further, they discuss the applicability of these techniques.

## 5. Analysis of SCTP and CMT Transport Protocols

In the following subsections, the main variants of SCTP and CMT-SCTP are presented.

### 5.1. SCTP and Its Variants

The SCTP specification was first proposed in RFC 2960 [68] in 2000, and it has since been updated in RFC 3309 [69] and RFC 4460 [70]. RFC 4960 [4] contains the current SCTP protocol definition, which IETF accepted in 2007. The SCTP is message-oriented, multi-streaming, and multi-homing, comprising a stable, connection-oriented, window-based CC service. The SCTP protocol allows a device to establish a logical connection across several interfaces, each with its own IP address. The SCTP additionally provides stream order and reliability, protects against SYN assaults, and requires selective ACK (SACK) usage. When the primary channel is unavailable due to congestion or connection failure, SCTP exploits the multi-streaming and multi-homing properties to improve the resilience of the communication network. Later, SCTP's major goals became load balancing and bandwidth aggregation. As a result, many protocol developers worked to improve the SCTP's load balancing and bandwidth aggregation performance.

A modified SCTP version, bandwidth aggregate-SCTP (BA-SCTP) [71], was introduced to aggregate the current bandwidth over several interfaces. However, in Westwood SCTP (W-SCTP) [72], an SCTP enhancement was proposed that aims at load balancing across multiple interfaces using bandwidth-aware scheduling. An extended SCTP, named load sharing SCTP (LS-SCTP) [73,74], was proposed to aggregate the bandwidth and maintain load balancing among multiple active transmission paths. Another extension of SCTP was proposed in [75] to mitigate the effect of packet loss in the lossy environment. It also limits the redundant data transmission over a different path to minimize the congestion in the network. The authors of [76] proposed a selective-redundancy multipath transfer (SRMT) strategy that employs the primary path to deliver data and the secondary path to transmit redundant data to prevent video data

quality loss. The partially reliable SCTP (PR-SCTP) is provided in [77]. Extra criteria are also provided for making the reliable SCTP adaptable for video streaming. The autonomous per path CC SCTP (IPCC-SCTP) approach [78] eliminates incorrect retransmission. It uses the idea of a unique route sequence number (RSN) for each path, which determines whether data packets are sent in an orderly or disorderly manner to each recipient. In [79], the authors advocated wireless multi-path SCTP (WiMP-SCTP), which aggregates bandwidth in the wireless environment by using data-stripping and data-duplicating modes for transmission. Shailendra et al. [47] recommended a competent MultiPath scheme for SCTP (MPSCTP) that simultaneously transmits data packets over multiple paths. MPSCTP is a solution for packet reordering and ineffective CWND growth, claiming greater throughput and retransmissions. However, Shailendra et al. [80] suggested a bandwidth estimation-based resource pooling (BERP) CC algorithm that enhances the throughput and performance of multipath communication. MPSCTP of [80] was later improved to include the data transmission rate on each lane, based on the complete path delay. Hence, such an approach minimizes the chunk latency on different channels, but because of its uniform bandwidth sharing strategy, it still has difficulties with available bandwidth utilization. To reduce the RBB in MPSCTP, Shailendra et al. [81] devised the Tx-CWND retransmission destination selection mechanism. Tx-CWND selects the path based on the size of the CWND. However, to reduce the processing overhead at the sender's level, Tx-CWND performs retransmissions on the identical underlying path. Notably, Tx-CWND performs path selection only when the path is marked inactive because of failure.

### 5.2. CMT-SCTP and Its Variants

Figure 5 demonstrates the internal architectural details considering the CMT-SCTP model. This model contains a multi-interface equipped transmitter (or source), multiple available interfaces underlying network paths, and a multi-interface equipped destination. On the transmitter side, the model contains four submodules: the Connection manager, Scheduler, fast re-transmitter, and flow control.

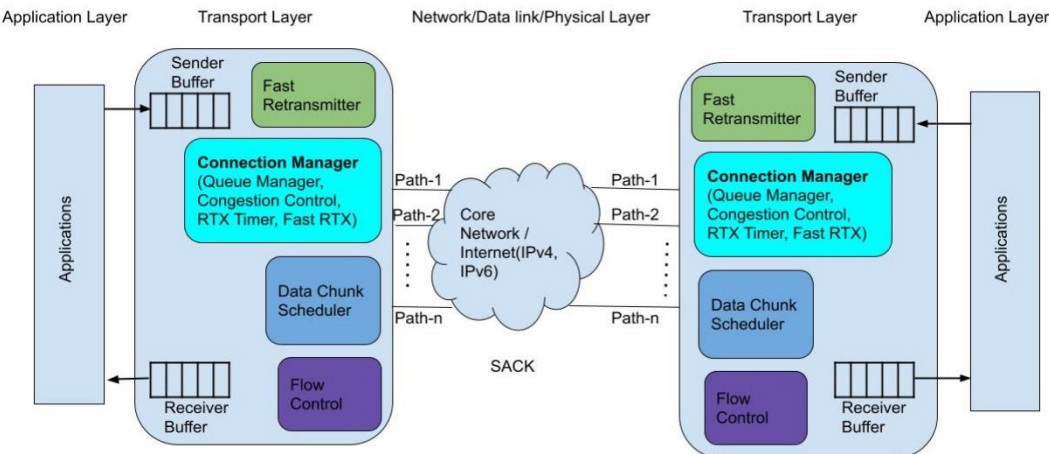

**Figure 5.** CMT protocol stack architecture.

Whenever the Layer-4-based multipath protocol obtains the data from the application layer, it creates chunks (data) out of it and keeps them in the sender buffer. After that, the scheduler picks up those chunks from the sender buffer and sends them through multiple interfaces to the underlying paths. However, this distribution policy depends entirely on the schedulers' design or distribution criteria. Finally, when all these chunks reach the destination, there is a high probability that they may be received out of order. To handle the disordered reception of these chunks, a buffer (of limited size) is maintained at the receiver side, which is usually called a receiver buffer. The main aim of

this buffer is to ensure the ordered delivery of data packets to the application layer. Then, after receiving data chunks, the receiver transmits a gap report to the transmitter using the concept of the SACK message.

CMT [5] sends data packets concurrently over multiple routes. Most CMT solutions improve the network's capacity utilization, robustness, throughput, and reliability. CMT schedules data chunks over many pathways using a round-robin mechanism. However, each path has a unique network feature (bandwidth and latency), and the CMT data scheduling mechanism does not consider path delay and bandwidth. As a result, the destination receives data not in any particular order. CMT reorders the received data packets using the receiver buffer's limited space. The receiver buffer becomes blocked as the number of unordered data packets increases because the destination does not pass the data packet to the upper layer until it receives the missing packet.

As mentioned earlier, the CMT retransmission issue was identified in [5], and the SFR algorithm was proposed as a solution. SFR keeps track of the receiver's highest transmission sequence number (TSN) ACK. The SFR algorithm considers that each data chunk in an association is atomic because it is assigned a unique TSN. It is noteworthy that TCP associates a sequence number with each data octet in the byte stream. SFR increases the CMT's performance in terms of data packet retransmission. Iyengar et al. [5] proposed another approach that keeps each destination's CWND distinct and allows it to expand independently. It utilizes a separate cumulative ACK (CUMACK) for each destination and modifies the CWND on the basis of the CUMACK received. CMT further reduces ACK traffic by postponing ACKs until at least two may be sent simultaneously. However, when it receives un-ordered data packets, CMT sends an immediate ACK. The reordering of ACK regularly increases because of frequent un-order data delivery. Delayed ACK for CMT (DAC) was also included in SFR to minimize ACK traffic [5]. To minimize the RBB problem, caused by the transmission of unordered data packets, Yilmaz et al. [43] introduced the NR-SACK approach to the free receiver-side buffer. The ACK technique ignores the segment, considering reordering and CWND growth.

Basic CMT is used in most concurrent systems, such as CMT-QA [40], distortion-aware CMT (CMT-DA) [82], and content-aware CMT (CMT-CA) [83]. Round-robin scheduling is used for data transmission among multiple paths in these CMT-based schemes. Each route, however, has a unique transmission latency and bandwidth. As a result, data packets arrive at the receiver in an unsorted state. CMT suffers from SACK overhead, undesired retransmission, receiver buffer blockage, excessive CWND reduction, and poor data chunk scheduling due to unordered data packet delivery [7,35]. Xu et al. [84] suggested a cross-layer fairness-driven CMT-SCTP that improves video data transmission and maintains fairness with competing flows. Xu et al. [40] also advocated a path quality-aware adaptive CMT (CMT-QA) scheduling approach for heterogeneous wireless networks. This scheme's main goal was to reduce out-of-order packet delivery by reducing unnecessary fast retransmission and reordering delays. The authors state that CMT-QA improves performance when multimedia data are transmitted across several pathways. However, as stated in [84], CMT-QA is problematic with fairness. The authors [45] enhanced CMT-SCTP in a heterogeneous network and proposed CMT-NC.

Arianpoo et al. [85] proposed a new network-coding-based CMT protocol (called SCTP-CMT) that makes use of Q-learning techniques [86]. Wu et al. [82] suggested a novel CMT-DA technique that improves video streaming quality in the wireless environment. This method mitigates the distortion by lowering the data packet loss rate of video streaming. Wu et al. [83] proposed the CMT-CA technique, which entails analyzing video contents to schedule data packets for better quality.

Dreibholz et al. [35] proposed a sender buffer splitting scheme that separates the sender buffer into numerous halves dependent on the number of various pathways. The authors claim that their method improves the RBB problem, but path inequality still causes local blocking.

The RBB [6,7] problem stemming from route failure was observed by Natarajan et al. [44], who proposed a novel condition, dubbed PF state, as a remedy. Due to network congestion or connection link failure, the PF state indicates that the target (destination) cannot be reached this way. Therefore, all new data packets are sent via the available alternate path.

Wu et al. [87] proposed the energy and goodput optimized CMT (EPOC) solution to deliver video streaming over multiple wireless paths. They presented two models; the first is an analytical framework to establish a relationship between energy consumption and goodput. The second model is a joint rate allocation scheme and FEC coding to reduce energy consumption with the required goodput. The authors claim that EPOC achieves better goodput, energy consumption, and distortion.

In [88], the authors presented loss-aware CMT (CMT-LA), which schedules packets according to packet loss and loss rate of every path to aggregate bandwidth and parallel transmission. The authors claim that CMT-LA reduces reordering delay and unnecessary retransmission more than the conventional CMT. Liu et al. [89] proposed a selective-retransmission-based CMT mechanism (CMT-SR) to improve the retransmission. CMT-SR constantly observes and investigates each path's delay and bandwidth and estimates the arrival time of packets. CMT-SR utilizes a pull-based and push-based mechanism to identify packet loss in time and prioritizes the retransmission. The experimental results indicate that the scheme achieves better quality and data delivery. Verma et al. [90] advocated an adaptive packet scheduling for CMT (A-CMT) that uses bandwidth and delay factors to identify the path situation and schedule data chunks accordingly. This method improves average performance in terms of throughput of the network system up to 13%. In [91], the authors proposed an RL-based CMT, which improves the fairness issue towards other subflows.

The functionality of a multipath transport protocol can be based on multi-homing, bandwidth aggregation, bandwidth estimation, path quality, the QoS of each subflow, buffer size and splitting, CWND updates, and so on. In addition, a CMT-SCTP protocol can function in a wireless, mobile, or general network. Furthermore, it can address various problems such as load balancing, packet reordering, RBB, and fairness.

Table 1 summarizes the key SCTP-based algorithms and approaches involved in multipath transmission.

**Table 1.** SCTP and CMT-SCTP transport protocols.

| Transport Protocol | Year | Based on | Network | Path | Problem to Address |
|---|---|---|---|---|---|
| SCTP [4,68,69] | 2000, 2002, 2007 | Multi-homing | General | General | Fault tolerance and resource aggregation |
| PR-SCTP [77] | 2004 | Bandwidth aggregation | General | General | Spurious retransmission |
| BA-SCTP [71] | 2004 | Bandwidth estimation | General | General | Scheduling, fairness |
| W-SCTP [72] | 2004 | Bandwidth estimation | General | Disjoint | Load balancing |
| LS-SCTP [73,74] | 2004 | Path quality | General | General | Spurious retransmission |
| m-SCTP [92,93] | 2005, 2007 | Soft handover | Mobile | General | Resource pooling |
| CMT-SCTP [5] | 2006 | Retransmission policies, CWND updates | General | Independent | CWND growth, retransmission |
| WiMP-SCTP [79] | 2007 | Aggressive failure detection | Wireless | Independent | Packet reordering |
| DAR-SCTP [94] | 2007 | Aggressive failure detection | General | Independent | Fault tolerance |
| cmpSCTP [95] | 2008 | Path quality | General | General | Packet reordering |
| mSCTP-CMT [96] | 2009 | Dwelling time, available bandwidth ratio, and RTT | Wireless | Disjoint | Packet reordering |
| CMT-PF [44] | 2009 | Aggressive failure detection | General | General | Retransmission, CWND growth |
| FPS-SCTP [97] | 2010 | Estimation of arrival times | Mobile | Disjoint | Packet reordering |
| WM2-SCTP [98] | 2010 | QoS of each subflow | Wireless | Disjoint | Resource pooling |
| Yilmaz et al. [43] | 2010 | NR-SACKs, Delay | General | General | Throughput |
| Dreibholz et al. [35] | 2010 | Buffer size and splitting | General | Asymmetric | Packet reordering, |

| | | | | | RBB problem |
|---|---|---|---|---|---|
| Adhari et al. [99] | 2011 | Optimized buffer handling | General | General | Packet reordering |
| Dreibholz et al. [100] | 2011 | Bandwidth estimation | General | Asymmetric | Resource pooling |
| CMT-BERP [101] | 2011 | Bandwidth estimation | Wireless | Asymmetric | Resource pooling |
| CMT-QA [40] | 2013 | Path's data handling capability | Wireless | Independent | Packet reordering, spurious retransmission |
| Cao et al. [102] | 2014 | CWND, Load sharing | Wireless | Asymmetric | Fairness, load sharing |
| DAPS [103] | 2014 | Round-trip time | Wireless | Asymmetric | RBB problem |
| Cao et al. [104] | 2014 | Receiver-based sending rate estimator | Wireless | Independent | Fault tolerance |
| Okamoto et al. [75] | 2014 | Bi-casting only important packets | Wireless | General | Spurious retransmission |
| CMT-DA [82] | 2015 | Utility maximization theory, path status estimation | Wireless | Independent | Throughput |
| Xu et al. [84] | 2015 | Path quality, window-based mechanism | Wireless | General | Fairness, packet reordering |
| CMT-CC [105] | 2015 | Cognitive approach | Wireless | General | Fairness, CWND growth |
| ENH-SCTP [106] | 2015 | CWND ranking | General | General | SCTP CC for LTE-A Network |
| MPSCTP [47,81] | 2011, 2013, 2015 | Additional sequence number, bandwidth estimation | General | Independent | Packet reordering |
| CMT-CQA [107] | 2015 | Quality of experience (QoE) path history information | Wireless | Asymmetric | CNWD growth, fault tolerance |
| CMT-CA [83] | 2016 | Markov decision process, feedback channel status | Wireless | Independent | CNWD growth |
| da Silva et al. [76] | 2016 | A secondary path is used to send redundant data | General | General | Spurious retransmission |
| CMT-NC [45] | 2016 | Network coding, group-based transmission | Wireless | Disjoint | Spurious retransmission, RBB problem |
| Arianpoo et al. [85] | 2016 | Q-learning and logistic regression | Wireless | Disjoint | Packet reordering, Receiver buffer blocking |
| A-CMT [90] | 2017 | Path delay and bandwidth | General | General | CWND growth |
| CMT-EA [87] | 2017 | FEC coding and rate allocation | Wireless | General | Energy conservation |
| CMT-SR [89] | 2017 | Bandwidth and delay | General | General | Spurious retransmission |
| Arianpoo et al. [91] | 2017 | Distributed Q-learning mechanism | Wireless | General | Fairness |
| Eklund et al. [108] | 2018 | Path characteristics, queuing status, and data flows | General | Independent | Queuing status and data flows |
| CMT-VR [109] | 2018 | Packet priority and rate less raptor coding | Wireless | General | Spurious retransmission |
| CMT-LA [88] | 2019 | Packet loss and loss variation | Wireless | General | Packet reordering, spurious retransmission |
| BRCPD [110] | 2019 | Buffer awareness, frame-level rate control | Wireless | General | Loss rate |
| CL-SCTP [111] | 2019 | Overdue messages, redundant frames | Wireless | General | Spurious retransmission |
| DAS [112] | 2021 | Delay aware | General | General | RBB problem, CWND growth |

Table 2 shows a categorization of SCTP and CMT-SCTP protocols according to the research challenges they address.

**Table 2.** A categorization of SCTP and CMT-SCTP transport protocols.

| SCTP and CMT-SCTP Variants | | | | | | | |
|---|---|---|---|---|---|---|---|
| Protocol | Retransmission | CWND Grown | Packet Reordering | RBB | Fault Tolerance | Resource Pooling | Packet Scheduling |
| WiMP-SCTP [79], MPSCTP [47,81] | | | ✓ | | | | |
| Dreibholz et al. [35] | | | ✓ | ✓ | | | |
| SCTP [4,68,69] | | | | | ✓ | ✓ | |
| LS-SCTP [73,74], Okamoto et al. [75], da Silva et al. [76], PR-SCTP [77] | ✓ | | | | | | |
| CMT-SCTP [5], CMT-PF [44] | ✓ | ✓ | | | | | |
| CMT-QA [40] | ✓ | | ✓ | | | | ✓ |
| CMT-NC [45] | ✓ | | | ✓ | | | |
| A-CMT [90], CMT-CA [83] | | ✓ | | | | | |
| Arianpoo et al. [85] | | | | ✓ | | | |
| CMT-LA [88] | ✓ | | ✓ | | | | |
| CMT-SR [89] | ✓ | | | | | | |
| m-SCTP [92,93], WM2-SCTP [98], CMT-SCTP [100], CMT-BERP [101] | | | | | | ✓ | |
| DAR-SCTP[94] | | | | | ✓ | | |
| cmpSCTP [95], mSCTP-CMT [96], PFS-SCTP [97], Adhari et al. [99] | | | ✓ | | | | |
| QoE-oriented [113] | | | | | | | ✓ |
| DAPS [103] | | | | ✓ | | | |
| CMT-CC [105], BRCPD [110] | | | | | ✓ | | |
| CMT-CC [105] | | ✓ | | | | | |
| CMT-CQA [107] | | | ✓ | | | ✓ | ✓ |
| Eklund et al. [108] | | | | | ✓ | | ✓ |
| CMT-VR [109], CL-SCTP [111] | ✓ | | | | | | |
| References [80,82,88,91] and References [114–121] | | | | | | | ✓ |

## 6. Analysis of MPTCP Protocols

In the following subsections, MPTCP is analyzed. Furthermore, MPTCP CC algorithms and multipath schedulers are considered.

### 6.1. MultiPath TCP (MPTCP)

MPTCP, like SCTP, is a standard connection-oriented protocol that enables multi-homing. The primary purpose of MPTCP is to disperse traffic among different routes. For numerous connections, MPTCP enables transparency at the application layer. In addition, MPTCP ultimately allows middle-box integration in today's Internet architecture [11,122–125]. MPTCP outperforms standard TCP and SCTP in the present Internet architecture, with data segments ripping in the middle of segments. As a result, MPTCP provides better deployment capabilities and performance. In MPTCP, a single tokenized session is divided into multiple subflows with an option to differentiate between MPTCP and TCP connections. Data packets use two different sequence numbers for lost packets detection and packet reordering at the receiver end.

The initial multipath transmission strategies [5,79,126,127] do not depend on CC; they are uncoupled with CC algorithms. However, the CC policy independently leads to the issue of low fairness during transmission. Khalili et al. [59] claimed that MPTCP is unconcerned with the best (Pareto) resource allocation strategy. Precisely, MPTCP does

not provide fairness during transmission as it allows the violation of the equilibrium of the resource allocation problem between multipath connections. Peng et al. [42] identified the fairness problem in multipath transmission and suggested the *fluid* concept for MPTCP algorithms to improve the system's stability and uniqueness.

An MPTCP multipath transmission technique can adapt the CWND of each subflow, if it is only coupled with CC. In this case, it can accomplish fairness. To solve the fairness problem, in [22,62], MPTCP uses an adaptive coupled CC policy by appropriately modifying the CWND growth policy with respect to the network status of each subflow. Furthermore, the linear systematic-coding-based MPTCP (SC-MPTCP) [128] implicates linked CC policy features that outperform MPTCP. However, these sophisticated coupled algorithms did not consider the real condition of the network (i.e., packet losses owing to congestion or noisy wireless channels [129–131]). Therefore, the network performance is degraded.

*6.2. MPTCP CC Algorithms*

In the present MPTCP implementation, a simple CC algorithm named linked-increases algorithm (LIA) [23]is used. LIA aims to improve throughput and move the data flows from a more congested to a less congested path. An MPTCP CC algorithm decides the CWND of each path and follows two phases to update the CWND of each path:

- *Slow-start*—the CWND escalates exponentially.
- *Congestion avoidance*(CA)—the CWND surges linearly.

In coupled CC algorithms, each subflow has information about other subflows while deciding its CWND that looks after the congestion in other subflows. By doing this, MPTCP instinctively can achieve more efficient load balancing, and MPTCP can become less aggressive [13]. Notably, the traditional LIA policies have severe problems such as TCP-friendliness, responsiveness, and load balancing. Khalili et al. [132] discovered the MPTCP protocol's fairness issue and the low channel utilization. For this reason, they designed an opportunistic LIA (OLIA) with respect to the CWND growth adjustment technique.

Traditional LIA policies could not deliver the optimum resource pooling and responsiveness that their strategy can. Peng et al. [133] proposed an efficient fluid-based model for a broad class of MPTCP algorithms that solve issues such as uniqueness, existence, and stability in the designing stage of the CC algorithm for MPTCP. Their method focused on TCP-friendliness, receptiveness, and CWND fluctuations as performance measures. Furthermore, the authors claim that they have designed a novel balanced link adaptation (BALIA) [42] technique to improve the performance of MPTCP CC policies. Their approach recognizes design standards that regulate appropriate receptiveness, window oscillation, and friendliness. On the other hand, these schemes do not take into account dynamic wireless channel properties and reduce the CWND size to half abruptly. This certainly impacts their throughput performance significantly in such an environment. In fact, Han et al. [134] also emphasized that the functionality of MPTCP is not completely flexible in wireless network scenarios. This is because MPTCP is completely based on a loss-based CC mechanism, which further hampers the protocol's performance due to excessive effort in managing multiple subflows.

An MPTCP protocol includes a bottleneck detection technique that determines subflows that share a point of congestion. Ferlin et al. [135] proposed a dynamic coupled CC algorithm (MPTCP-SBD) with a concrete shared bottleneck detection technique for MPTCP. On the basis of the shared/non-shared bottleneck link identification approach, their CC algorithm dynamically couples/decouples subflows. Lubna et al. [136] proposed the dynamic OLIA (D-OLIA) scheme, which adapts the CWND size while exploiting the current delay of the available paths. The current paths' delay estimation in D-OLIA is entirely based on estimating changes in the RTTs.

The current policies of CC algorithms generally recognize packet loss as a sign of congestion. This way, random packet loss may be mistaken for congestion packet loss. In addition, there is no corresponding CWND adjustment method for different packet losses. Therefore, a blind reduction of CWND for traffic control will only lead to the deterioration of MPTCP performance. In this context, Cai et al. [137] proposed a packet loss differentiation-based OLIA, called D-OLIA. D-OLIA can determine the type of packet loss by combining the eigenvalues of delay jitter and CWND jitter to make up for the deficiency of judging only by delay or CWND. Finally, Abdrabou and Prakash [138] analyzed the MPTCP multi-homed wireless interfaces, where one interface is associated with a Wi-Fi network, and the other emulates a 3G or 4G link. The authors suggested that the MPTCP coupled CC algorithm can transfer more data on the Wi-Fi link than the dedicated link at a limited receiver buffer.

MPTCP Latency Reduction CC Algorithms

The implications of multipath scheduling on delay-sensitive applications (e.g., live-streaming, gaming, and video conferencing) were thoroughly examined by Yedugundla et al. [139]. In [30], the authors looked into the technicalities of routing over many pathways and traffic splitting issues. Furthermore, they provided in-depth information on improving network performance by implementing multipath technology across networks. However, the authors of [67] presented a comprehensive review of the challenges of multipath traffic splitting in terms of layer characteristics. Cao et al. [140] found that existing multipath algorithms achieve only coarse-grained load balancing of congestion status using packet losses and suggest a solution based on the congestion equality principle. They created a queuing delay parameter-based system that controls congestion using packet queuing delay to accomplish fine-grained load balancing.

Oh and Lee [141] proposed a unique MPTCP path delay and receiver buffer scheduling policy. This technique estimates unordered packets based on individual subflow performance differences and assigns packets to each subflow. The trade-off between network throughput and delay performance is efficiently adjusted using this approach. Xu et al. [142] introduced pipeline network coding-based MPTCP (MPTCP-PNC) to decrease needless coding delays and bandwidth overuse in the present coding-based system. MPTCP-PNC employs new economical coding, quality-based delivery planning, and associated data transmission policies to increase overall system performance. Cui et al. [41] proposed FMTCP to reduce the reliance on individual subflow in multipath transmission while ignoring the influence of a poor-performing subflow on other subflows. FMTCP uses fountain codes to handle numerous pathways' varied features efficiently. In terms of goodput, FMTCP beats MPTCP when the pathways have a variety of delays and losses, as well as reduced delay and jitter. For multipath and multisource transmission, Thomas et al. [143] presented a hybrid CC strategy that results in larger bandwidth aggregation than conventional MPTCP. In their framework, a smart module (inside the network) dynamically examines the information concerning the network topology. Their CC technique uses a multi-flow CC policy with network assistance (MFCCNA). The ultimate goal of MFCCNA is to increase network resource usage while maintaining network friendliness. MFCCNA continues to operate without taking into account the harshness of the window expansion policy. As a result, data packet loss and performance degradation occur.

NC-MPTCP [48] was proposed by Li et al. to reduce retransmission in the event of a delay. However, to avoid unnecessary fast retransmission, the SC-MPTCP uses redundant code [128]. Finally, Zhou et al. [144] developed the CWND adaptation MPTCP (CWA-MPTCP) to control the transmission rate for each sub-flow to have a similar end-to-end latency.

Lee et al. [145] showed concerns about the performance (high-speed low latency) of MPTCP-based CC schemes in a 5G network scenario. The authors showed that the currently implemented MPTCP-based CC schemes (e.g., LIA, OLIA, and BALIA) fail in

attaining lower end-to-end delay. These schemes cannot perform well when the condition of the millimeter-wave (mmWave) link fluctuates frequently from line-of-sight (LoS) to non-LoS (NLoS) and vice versa. In the case of mmWave 5G networks, these impulsive changes from LoS to NLoS and vice versa can cause severe throughput degradation because of frequent packet losses. Nevertheless, to deal with such a situation, the underlying link-layer/medium access control (MAC) sub-layer performs frequent MAC-level retransmissions that hide such packet loss events to the MPTCP-based CC schemes at Layer-4. Hence, they do not adapt (reduce) their CWND growth unnecessarily. Although through this notion, these MPTCP-based CC schemes somehow saved the throughput performance from degrading, MPTCP is still unaware of the exact condition of the wireless channel capacity. Hence, this leads to the issue of buffer-bloating, and it is well known that this issue becomes quite severe in the case of 5G networks. In addition to this, Lee et al. [145] showed that weighted Vegas (wVegas) [140] does not utilize the channel capacity appropriately when it competes with packet loss-based MPTCP-based CC schemes. The wVegas continuously adapts its CWND growth on the basis of queueing delay estimations. Actually, the loss-based MPTCP-based CC schemes try to rapidly utilize the available channel capacity (i.e., aggressive CWND growth) until a loss event occurs in the network. Moreover, when these schemes attempt to fill a queue (bottleneck), wVegas, as per its policy, tries to adapt (lessen) its CWND growth and minimize the bottleneck queue length. Although delay-based (wVegas, CL-ADSP [146], and A-DSP [147]) MPTCP-based CC schemes are less aggressive in utilizing channel capacity than the other loss-based MPTCP-based CC schemes, they are far better than loss-based MPTCP-based CC schemes in attaining low latency and high speed. This is because the delay-based, MPTCP-based CC schemes can control and assist in minimizing the bottleneck queue delay accurately, while the loss-based MPTCP-based CC schemes can aggravate the buffer-bloating problem in the system. This motivated the authors [145] to design a delay-based MPTCP-based CC scheme, called delay-equalized FAST (DEFT). The authors utilized and incorporated the concept of adaptive CWND reduction policy of FAST-TCP [148] in DEFT to achieve improved throughput performance. Moreover, DEFT incorporates a delay-equalizing algorithm that minimizes additional reordering delay in the receiver buffer. Furthermore, DEFT takes care of the dynamic properties of 5G mmWave (links) networks. The main objective of DEFT is to sustain fast responsiveness corresponding to each subflow by retaining a certain number of backlogged packets.

In the Internet of vehicles (IoV) networks, the traffic is delay-sensitive. Therefore, MPTCP algorithms must be proposed for satisfying delay constraints and providing reliable communication over such heterogeneous lossy IoV networks. Pokhrel and Choi [149] showed concerns about the performance issues with MPTCP in delay-constraint applications. Subsequently, the authors suggested a delay-sensitive MPTCP-based scheduler (for the case of the IoV) that aims to improve the load balancing procedure and ultimately improve the throughput performance of the protocol.

### 6.3. Multipath Schedulers

In multipath communication, three problems must be addressed: (1) the delivery of unordered data at the receiver side; (2) the different priorities that data packets have (data packets are generated by diverse media streams such as audio, video, graphics, and text); and (3) the low utilization of the available multipath bandwidth. To minimize the first two problems, Huang et al. [116] suggested the adaptive ordering predicting scheduling (AOPS) scheme that uses the packet arrival order to adapt the transmission rate of each path. However, the AOPS scheduler is a part of a Multipath datagram CC protocol, MP-DCCP, used for multimedia streaming. Notably, MP-DCCP [116] is an unreliable Layer-4 protocol with a CC procedure that includes AOPS.

The paths in a 5G network have dynamically changing channel characteristics. For example, the delay characteristics of a Wi-Fi network in a public hospital can vary over

time due to different numbers of users. These delay characteristics generate different congestion levels and thus dissimilar dynamicity levels during the night. Due to the paths' heterogeneity, sent packets arrive at the destination out of order, leading to HoL blocking, ultimately reducing the performance. Consequently, the design of a multipath scheduler requires deploying a policy that will cope with the heterogeneous characteristics of paths (e.g., delay and loss).

A multipath scheduler can address the HoL blocking problem by deciding to wait for improved traffic conditions to appear for the transmission of a packet. To this end, the blocking estimation-based MPTCP scheduler (BLEST) [150] and the earliest completion first (ECF) [151] schedulers introduce a *wait* action. However, both schedulers were not designed for dynamically path-changing characteristics. Notably, in real-time multimedia networks, the path dynamicity level (i.e., path delay variation and packet loss rate) can vary extensively over time.

Peekaboo [152] is an adaptive, lightweight multipath scheduler that was implemented in MPQUIC. It is an RL-based scheduler that considers paths' dissimilar characteristics. Peekaboo forces an online learning mechanism and combines a stochastic adjustment policy to adapt to the dynamic characteristics of the paths. Peekaboo was suggested considering the scheduler's dynamic adaptation to the rapidly changing channel conditions. Peekaboo works in two folds:

(1) The *Deterministic* approach; and
(2) The *Stochastic tuning* approach.

In the Deterministic approach, Peekaboo selects a path or waits for the availability of a better path for the transmission. It then applies the Stochastic tuning approach on top of the Deterministic approach to efficiently handle the dynamicity level over a selected path. Peekaboo offers superior performance to the other compared schedulers. Peekaboo's performance improvements reached 36.3% in scenarios of a real-time multimedia network.

Recently, a new RL-based scheduler has been designed and implemented by Zhang et al. [153], considering the issue of path heterogeneity. Their proposed reward factor takes delay, packet loss, and throughput into consideration while performing data scheduling.

The modified-Peekaboo (M-Peekaboo) [154] builds on Peekaboo and extends its usage to 5G networks. In particular, M-Peekaboo extends Peekaboo's learning scheme for path selection toward 5G scenarios that may include paths operating on different frequencies such as mid-band and mmWave. The results illustrate the benefits of employing a learning-based multipath scheduler for 5G networks and motivate further studies of advanced learning schemes that can adapt quickly to the path conditions and take into account the emerging features and requirements of 5G and beyond networks.

Amend et al. [155] assessed the restrictions of existing schedulers when video-on-demand traffic is transferred in multipath scenarios. They introduced a new scheduling algorithm called cost-optimized multipath (COM). COM decreases the cost of mobile network operators while delivering video-on-demand traffic over multipath network access. The authors showed preliminary testbed results, demonstrating the cost benefits of the COM algorithm. Further, they proved that the correct balance could be obtained for the video traffic between the operator cost and the user QoE.

The stochastic optimal scheduler for MPTCP (SOS-MPTCP) [156] uses the Lyapunov optimization technique in a software-defined wireless network (SDWN). A Lyapunov optimization technique employs a Lyapunov function to control a dynamical system to achieve stability optimally. SOS-MPTCP tries to maximize the throughput and minimize the price cost for users. In SOS-MPTCP, the trade-off problem between the performance and price cost for users is addressed. In the proposed SDWN architecture, the controller provides status information for each path back to mobile terminals. As a result, SOS-MPTCP can decide on (1) packet admission control; (2) packet distribution control;

and (3) data traffic purchasing control. The efficiency of the proposed trade-off optimization of this scheduler was proved through experimental results.

During data transmission, MPTCP enables capacity aggregation and flexibility. However, data transmission across several pathways does not consider the path characteristic (delay and capacity) to be a relevant influence [122,157]. Because of this, the destination receives uncontrolled data packets that cause CWND reduction and unnecessary retransmission. To overcome unordered packet delivery, Le and Bui [158] proposed an MPTCP forward delay-based packet scheduling (FDPS), while Yang and Amer [159] proposed a one-way delay-based MPTCP scheduler. Both policies distribute packets over different subflows on the basis of the predicted route delay of each source. On the other hand, Ni et al. [160] suggested an offset compensation-based packet scheduling (OCPS) that schedules on the basis of the SACK feedback information.

Initially, Kelly and Voice [161], Han et al. [162], and Wang et al. [163] suggested some schemes that only utilize the ideal paths accessible to users and are optimal in static configurations where paths have comparable delay variations (concerning RTTs). Earlier schemes [161–163] only utilize the ideal paths accessible to users and are optimal in static configurations where paths have comparable delay variations (concerning RTTs). Nevertheless, these schemes occasionally fail to adapt to situations where paths have a higher loss probability. These schemes cannot rapidly detect available channel capacity because they do not probe such kinds of paths capably. Moreover, these schemes occasionally exhibit flappiness in the system. In the case of several available good paths to a user, a user tries to flip its traffic load onto these available network paths arbitrarily (i.e., a simple case of uncertainty in data scheduling) [59,126,164]. Hence, due to such issues, the MPTCP CC policy does not comply with these fairly static kinds of schemes [161–163] directly. In fact, the importance of self-adaptation has been given in the design of the MPTCP CC policy. The plain notions of MPTCP CC are as follows:

1. *Do not harm:* This notion states that an MPTCP flow (connection) should not consume any more channel capacity on any of the underlying paths than a single-path TCP flow (connection). An MPTCP flow should be fairer towards other competing TCP flows and should share the channel capacity equally.
2. *Improve throughput*: This notion states that an MPTCP flow (connection) should perform better in terms of throughput than a single-path TCP flow (connection).
3. *Balance congestion*, as suggested by Raiciu et al. [23,165]:This notion states that an MPTCP CC policy's scheduler should transfer as much traffic load as possible from its highly congested paths onto its least congested paths, provided that the first two notions are met. Hence, MPTCP utilizes the available bandwidth effectively as compared to single-path TCP.

Moreover, MPTCP also suggests improved throughput and fairness performance than a single-path TCP. Apart from this, MPTCP successfully manages the issue of flappiness in the system. However, Khalili et al. [59,166]performed extensive analysis and simulation measurements and exhibited that the MPTCP CC design policy (LIA) cannot entirely satisfy balanced congestion (notion 3). The authors advocated that MPTCP is not Pareto-optimal, particularly when an MPTCP connection shares a bottleneck link with a single path TCP connection. They demonstrated that an MPTCP flow (connection) can be exceedingly unfair towards another competing TCP flow (connection) over highly congested network paths. The authors suggested that LIA fails to simultaneously offer the responsiveness and optimum resource pooling. Subsequently, the authors improved the LIA design and proposed an OLIA scheme for MPTCP. The main aim of OLIA is to provide responsiveness and optimum resource pooling at the same time. Similarly, in the past, several authors have suggested different MPTCP-based CC schemes, such as BALIA, A-DSP, CL-ADSP, wVegas, adapted-OLIA [167], and Couple+ [168], which inherently utilize the above-mentioned notions. Nevertheless, while designing all these schemes, the researchers gave the ultimate

priority to network fairness. In fact, all these schemes ensure that no MPTCP connection should achieve better overall throughput performance than that of other competing single-path TCP connections on the best available end-to-end network path, no matter whether MPTCP connections share a bottleneck or not. Although to ensure fairness, it is true that by following this concept, the throughput performance of all the flows will be almost comparable, and no flow will harm any other flow. Still, it is unfair towards MPTCP connections. Wei et al. [169] deeply discussed and revealed such issues and showed that the conventional fairness-based MPTCP CC policy highly restricts the subflows' throughput performance during the case of non-shared-bottleneck-links, and such policy only attains the comparable performance as traditional TCP. Moreover, the authors have shown that the previously suggested schedulers, such as DPSAF [170] and BLEST, consider each subflow as an independent single TCP, and they did not contemplate that all subflows' CWND should be treated as coupled, during the CA phase to guarantee fairness. Subsequently, Wei et al. [169] suggested and implemented the shared bottleneck-based-CC (SB-CC) scheme. The authors' primary objective is to leverage the explicit congestion notification (ECN) policy to uncover subflows' information sharing single or multiple bottleneck links in an MPTCP flow (connection), and subsequently, the scheme perceived shared bottleneck sets (SBS). Moreover, to satisfy notion 2 (see above discussion), the SB-CC-based-coupled CC scheme has effectively estimated the congestion level corresponding to each subflow, keeping in mind the bottleneck fairness and optimum load balancing. Consequently, the scheme utilizes it for efficient CWND adaptations. Further, the authors provided another scheme called SB-based forward prediction scheduling (SB-FPS) for MPTCP. In SB-FPS, the authors predict the upcoming behavior for each subflow by taking SBS into consideration and schedule transmissions accordingly.

Sathyanarayana et al. [171] revealed the performance issues of MPTCP while dealing with several LTE networks. The authors showed concerns towards MPTCP's CWND adaptation scheme when it operates on rapidly changing network conditions. It may also be the case that the performance of single-path TCP may be better than that of MPTCP policy. Hence, the intelligent eradication of some low-quality paths and the addition of some good-quality paths for transmission is an essential requirement in the case of MPTCP. Further, there is the possibility that MPTCP likely suffers from the issue of buffer-bloating at the wireless links (cellular). Consequently, the authors have suggested a client-based MPTCP (cMP-TCP), aiming to enhance the MPTCP's throughput performance from the client side. cMP-TCP seeks to detect the bottlenecks more precisely over an end-to-end MPTCP connection. Hence, cMP-TCP accordingly selects the best available network paths for transmission.

### 6.3.1. Energy-Efficient Multipath Schedulers

With the advent of resource constraint devices, the requirement of energy-efficient multipath communication becomes necessary. To this end, Wu et al. [53,154] proposed an energy-aware and priority frame-based packet scheduler for the heterogeneous wireless environment to enhance the goodput, delay, and energy consumption in multipath transmission. Wang et al. [172] presented an MPTCP variant based on a genetic algorithm, a rate distribution vector, and an energy-aware scheduler to optimize devices' throughput and energy consumption in a heterogeneous wireless network. In another work, Zhao et al. [173] proposed a new MPTCP algorithm by minimizing the flow completion time of transmission to minimize the system's energy consumption in the cloud-based data center. Trinh et al. [174] proposed a low-energy-consumption-path-based scheduling to enhance the throughput and energy consumption of the system in the wireless environment. Morawski and Ignaciuk [175] also advocated the concept of minimizing energy consumption during multipath scheduling. They suggested their approach on the basis of the available three default schedulers (i.e., default, round-robin, and redundant) originally designed for MPTCP.

Their suggested proposal combines the assistances of default and redundant schedulers of MPTCP. Along with that, their ultimate motivation is to reduce energy consumption during multipath bidirectional communication. The authors show that the redundant scheduler is highly greedy and unfair concerning network level utilization and energy dissipation. Peng et al. [176] also addressed the issue of higher energy consumption when a smart mobile device uses multiple interfaces for concurrent transmissions. The authors suggested the improved MPTCP algorithms for such devices by jointly addressing the tradeoff between throughput performance and energy dissipation. They suggested two mechanisms for both fixed size transfer (data) and real-time applications. Their simulation results suggest that the energy consumption is reduced to 22% without compromising the throughput performance as compared with the default algorithms of MPTCP. Further, Lim et al. [177] planned, implemented, and assessed an energy-aware MPTCP (eMPTCP) whose ultimate aim was to reduce energy consumption while not compromising the performance. eMPTCP utilizes the combination of deferred subflow formation and energy-aware subflow management to achieve their thought. Their experimental results show that eMPTCP suggests around 90% and 50% reduction in energy consumption for small and large file downloads, respectively.

Morawski and Ignaciuk [178] considered the issue of optimal traffic load dissemination amongst available interfaces underlying network paths with an overall objective of minimizing energy consumption. Hence, the suggested scheme is well suited for low-powered battery devices equipped with more than one interface. For this, they considered two scenarios: (1) uncoupled transmission channels, and (2) coupled transmission channels. For these scenarios, they proposed two dynamic MPTCP scheduling allocation algorithms. The authors revealed that in the case of the second scenario, the proportion of energy consumption had dropped substantially without loss of throughput performance. Considering the same objective of improving performance concerning energy consumption without compromising throughput performance, Dong et al. [179] suggested the energy-saving scheduler (ES-MPTCP). ES-MPTCP offers an optimized function (target) that estimates the exactness of a number of subflows utilized for transmission, saving energy consumption without compromising the throughput performance.

### 6.3.2. Optimization of MPTCP Schedulers for Real-Time Applications

From the start, MPTCP was designed to provide fully organized and dependable connection-oriented services. Despite this, MPTCP is insecure for real-time applications due to this characteristic. To support real-time applications (e.g., video streaming), Xu et al. [180] proposed a partially reliable MPTCP (PR-MPTCP). Several other researchers contributed to MPTCP optimization for real-time applications, while significant MPTCP variants were introduced: QoS-MPTCP [181,182], message-oriented MPTCP (MO-MPTCP) [183], quality-driven multipath TCP (ADMIT) [184], and cross-layer scheduler [185]. The authors of [186] proposed scheduling according to packets arrival time and controlled sending of retransmission and claim it reduces packet ordering and delays issues to improve throughput. In [187], the authors proposed a loss-aware throughput estimation (LATE) scheduling algorithm for MPTCP that achieves better goodput. The authors of [188] outlined an adaptive subflow management scheme for the wireless environment, and the simulated result reveals that it achieves better results in terms of throughput. To mitigate reordering and the RBB issue, a bandwidth exploitation approach, adaptive and efficient packet scheduler (AEPS), is presented in [189]. In another work, the authors of [190] presented the optimal utilization of traffic flow to mitigate packet loss and improve throughput.

In MPTCP protocols, the number of transmissions (and thus MPTCP delay) can be reduced using opportunistic routing (OR) [191]. OR algorithms exploit the broadcast nature of the wireless channels. By using the broadcasting technique, these routing algorithms can improve network performance and enhance data transmission's delivery

rate and reliability [192]. In OR, the next hop forwarders (network nodes) are not fixed. Aljubayri et al. [191] adapted OR on some MPTCP protocols (i.e., conventional MPTCP, multipath TCP traffic splitting control (MPTCP-TSC) [193], and redundant MPTCP (Re MPTCP) [194]) in an IoT environment. They achieved a reduction in MPTCP delay. Their outcomes demonstrate that such OR-based policies can perform better than existing policies. However, it is noteworthy that even though OR can improve network performance, network resources are allocated to the flows, despite their QoS needs.

The HoL blocking problem (caused when a buffer size is limited) can degrade the performance of the MPTCP. To solve this problem, Choi et al. [195] suggested an optimal load-balancing scheduler considering the degraded throughput performance issue of the default LINUX-based MP-TCP scheduler. The authors derived and modelled the theoretical limit of the scheduler's attainable throughput performance for the available buffer size.

Table 3 summarizes key multipath schedulers, while Table 4 summarizes MPTCP transport protocols.

**Table 3.** Summary of multipath schedulers.

| Scheme | Year | Scheduling Policy | Description |
|---|---|---|---|
| [48,49,107] | 2013, 2017,2015 | Quality-aware | More traffic amount is scheduled to a path in which the quality coefficient is higher. Traffic is continuously monitored and estimated based on different QoS parameters of each path. Such QoS parameters are loss rate, transmission rate, congestion, capacity, etc. |
| [31,52,103,114,158,159,166,196] | 2014,2017,2017,2019, 2014,2013,2012,2017, 2018 | Delay-based | Scheduling of data packets is carefully decided based on the data transmission delay of each path and the most widely used scheduling policy. |
| [115,160] | 2015,2016 | Feedback | Scheduling of data packets is based on feedback information from SACKs to decide previous scheduling performance and future scheduling options. |
| [116] | 2015 | Packet order prediction | Data packet scheduling at the sender is decided based on a prediction of packets arrival order at the receiver. |
| [195,197,198] | 2017,2019,2017 | Load balancing | The scheduler maintains the load balance of data packets flow between each subflow of multipath transmission. |
| [108,141,199] | 2015, 2018,2017 | Queuing status | The scheduler decides data scheduling based on queuing delay of individual subflow. |
| [38,80,117] | 2013,2017,2016 | Bandwidth-aware | Data packets scheduling is based on the available bandwidth of each path. |
| [82,88,119,200,201] | 2017,2015,2019,2014 2014, 2021 | Loss-aware | Data packets are scheduled according to packet loss and variations in loss across the available multiple paths. |
| [118,120,121] | 2017,2016,2017 | Energy-aware | Data packets are scheduled according to the energy consumption of the available multiple paths. |
| [90,202] | 2017,2017 | Hybrid | Multiple scheduling criteria are taken into account. |
| [203] | 2018 | Priorities-aware | Data packets of high priority are scheduled for high-quality interface links. Priority depends on the application at hand. |
| [204] | 2022 | Throughput ratio-based scheduling | The scheme maintains the packet assignment ratio to the two subflows equivalent to the throughput ratio of the considered two subflows. |
| [205] | 2022 | Path rank-based | Individual path rank is computed based on the successful transmission rate, and data chunks are allocated accordingly. |

**Table 4.** MPTCP transport protocols.

| Transport Protocol | Year | Based on | Network | Path | Problem to Address |
|---|---|---|---|---|---|
| MPTCP [8,9] | 2011–2013 | Simultaneous transmission over multiple subflows | General | Disjoint | Bandwidth aggregation |
| NC-MPTCP [48] | 2012 | Network coding, compensating the lost packets | General | General | RBB |
| Hassayoun et al. [206] | 2012 | Retransmission | General | General | Packet reordering |
| QoS-MPTCP [182] | 2012 | Partial reliability | General | General | Network availability and QoS |
| Peng et al. [133] | 2013 | Fairly allocation of bandwidth | General | General | Fairness, resource pooling |
| Khalili et al. [59] | 2013 | Optimal resource pooling and responsiveness | General | General | Pareto-optimality |
| Coudron et al. [207] | 2013 | Opportunistic linked increases | Cloud | Independent | Pareto-optimality |
| Van der Pol et al. | 2013 | Simultaneous use of multiple paths | Open | General | Link failure |

| | | | | | |
|---|---|---|---|---|---|
| [208] | | | Flow | | |
| A-MPTCP [209] | 2013 | Additional subflow creation mechanism | CloudNet | General | Transmission delay |
| CWA-MPTCP [144] | 2013 | End-to-end path delay | Wireless | Independent | RBB |
| SC-MPTCP [128] | 2013–2014 | Linear systematic coding | General | General | Retransmissions, RBB |
| Yang and Amer [159] | 2014 | In order arrival scheduling | General | General | RBB |
| FMTCP [41] | 2015 | Fountain-code-based | General | Disjoint | Higher total goodput, lower delay |
| Ni et al. [160] | 2015 | Feedback information from SACK | General | Independent | RBB, enhanced throughput |
| Le and Bui [158] | 2015 | Forward-delay-based packet scheduling | General | General | RBB, enhanced throughput |
| AMTCP [210] | 2015 | Addition of a dynamic number of the subflows | Data center | General | Throughput |
| Ferlin et al. [135] | 2016 | Shared bottleneck detection | General | General | Fairness, throughput |
| Wu et al. [121] | 2016 | Energy-aware and prioritize frame scheduling | Wireless | Independent | Goodput, delay, energy consumption |
| Xu et al. [142] | 2016 | Pipeline network coding | Wireless | General | Delay, bandwidth utilization |
| Oh and Lee [211] | 2016 | Feedback-based path failure detection | General | General | Retransmissions, RBB |
| Wu et al. [117] | 2016 | Priority-aware scheduling and FEC | Wireless | General | End-to-end delay, bandwidth utilization, and goodput |
| Cao et al. [212] | 2016 | Receiver-centric buffer blocking-aware data scheduling | Wireless | Asymmetric | Reordering, RBB |
| Mmptcp [213] | 2019 | Randomizing of a flow's packets | Data center | Independent | Loss rate, throughput |
| Cui et al.[214] | 2016 | End-to-end coding | General | General | Throughput and latency |
| Xue et al. [168] | 2016 | Network coding, end-to-end CC | Wired, wireless | General | Fairness |
| Choi et al. [195] | 2017 | Optimal load balancing scheduler | Wireless | General | HoL blocking, throughput |
| Wang et al. [172] | 2017 | Genetic algorithm, a rate distribution vector, energy-aware Scheduling | Wireless | General | Throughput, energy consumption |
| Kimura et al. [202] | 2017 | Highest sending rate, largest window space, and lowest time/space-based scheduling | General | General | Throughput |
| Lim et al. [151] | 2017 | Earliest completion first scheduling | General | Asymmetric | Bandwidth aggregation |
| BELIA [38] | 2017 | Estimation of the real bandwidth of the link | General | General | Throughput |
| Lin et al. [215] | 2018 | Packets retransmission | General | General | Data latency |
| Le and Bui [158] | 2018 | Forward delay-based packet scheduling | General | Asymmetric | Reordering |
| Ferlin et al. [216] | 2018 | FEC | General | Asymmetric | Retransmissions, HoL blocking |
| Wu et al. [217] | 2018 | Delay–energy–quality-aware | Wireless | Asymmetric | Throughput |
| Mena et al. [218] | 2018 | Capacity estimation of path | Wireless | Independent | Handover |
| Zhu et al. [219] | 2018 | Bottleneck bandwidth and round-trip propagation Time | Wireless | General | Fairness |
| Morawski et al. [178] | 2018 | Optimal load distribution | Wireless | General | Energy consumption |
| Elgabli et al. [220] | 2018 | Scalable video coding | Wireless | Independent | Fairness |
| Zhao et al. [173] | 2019 | Minimizing the flow completion time | Data center | General | Energy consumption |
| Trinh et al. [174] | 2019 | Low energy consumption paths to deliver data | Wireless | General | Throughput and energy efficiency |
| Könsgen et al. [221] | 2019 | Allocation of link capacity using mixed linear programming | General | General | Throughput and fairness |
| Pang et al. [222] | 2019 | Queuing cache balance factor | Data center | General | Bandwidth aggregation, load balancing |
| Li et al. [223] | 2019 | Reinforcement learning | Wireless | General | Aggregate throughput |
| Hurtig et al. [224] | 2019 | Block estimation and the shortest transmission time first scheduler | Wireless | Asymmetric | Transmission times |
| CL-ADSP [146] | 2019 | Delay-variation-based adaptive fast retransmission policy | Wireless | Asymmetric | RBB problem and unnecessary retransmissions |
| Dong et al. [60] | 2019 | Loss and delay-based scheduling | Wireless | General | Bandwidth consumption |
| Shi et al.[225] | 2019 | A load-balancing mechanism based on congestion feedback | General | Asymmetric | Delay and stability |
| Thomas et al. [226] | 2020 | Normalizing the growth of | General | General | Fairness |

| | | individual subflow | | | |
|---|---|---|---|---|---|
| Hwang and Yoo [227] | 2020 | Multi-homing features of low memory devices | Wireless | General | RBB, memory |
| OLS [186] | 2021 | Using the latency of the path, out of order packets | Mobile | General | Throughput, reordering |
| MPCOA [54] | 2021 | Using congestion, buffer, bandwidth | General | General | Throughput, resource, RBB |
| LATE [187] | 2021 | Loss aware | Wireless | General | Goodput, latency |
| Li et al. [188] | 2021 | Number of subflows | Wireless | General | Throughput |
| AEPS [189] | 2021 | By exploiting the bandwidth | General | General | Reordering, RBB |
| MFVL HCCA [190] | 2021 | The exploitation of traffic flow | Wireless | Wireless | Packet loss, goodput |
| ES-MPTCP [179] | 2022 | Optimization through energy consumption | General | General | Throughput, energy consumption |

Table 5 shows a categorization of MPTCP protocols according to the research challenges they address.

**Table 5.** A categorization of MPTCP protocols.

| Protocol | ns/mi | irm/es/u/h | Ag | BBR/H | ro | ug/hn/et | Sc/he/du/la |
|---|:---:|:---:|:---:|:---:|:---:|:---:|:---:|
| SC-MPTCP [128] | ✓ | | | | | | |
| Oh and Lee [211] | ✓ | | | | | | |
| CL-ADSP [146] | | | ✓ | | ✓ | | ✓ |
| A-DSP [147] | | | ✓ | | ✓ | | ✓ |
| Peng et al. [133] | | ✓ | | | | | |
| Ferlin et al. [135] | | ✓ | | | ✓ | | |
| Kaiping et al. [168] | | ✓ | | | | | |
| Zhu et al. [219] | | ✓ | | | | | |
| Elgabli et al. [220] | | ✓ | | | | | |
| Konsgen et al. [221] | | ✓ | | | ✓ | | |
| MPTCP [8,122] | | | ✓ | | | | |
| Xu et al. [142] | | | ✓ | | | | ✓ |
| Wu et al. [117] | | | ✓ | | ✓ | | |
| NC-MPTCP [48] | | | | ✓ | | | |
| Ni et al. [160] | | | | ✓ | ✓ | | |
| Yang and Amer [159] | | | | ✓ | | ✓ | |
| Lim et al. [151] | ✓ | | ✓ | | | | |
| A-MPTCP [209][210] | | | ✓ | | | | ✓ |
| Ferlin et al. [217] | ✓ | | | ✓ | | | |
| Thomas et al. [226] | | ✓ | | | | | ✓ |
| Pang et al. [222] | ✓ | | ✓ | | | | |
| Dong et al. [60] | | | | | ✓ | ✓ | |
| CWA-MPTCP [144] | | | | ✓ | | | |
| Le and Bui [158] | | | | ✓ | ✓ | ✓ | ✓ |
| Cao et al. [212] | | | | ✓ | | | |
| Choi et al. [195] | | ✓ | ✓ | | | ✓ | |
| AMTCP [209] | | | | ✓ | | | |
| Jiyan et al. [121] | | | | ✓ | | | ✓ |
| Mmtcp [213] | | | | ✓ | | | |
| Cui et al. [214] | | | | ✓ | | | ✓ |
| Wang et al. [172] | | | ✓ | ✓ | | ✓ | |
| Kimaura et al. [202] | ✓ | | ✓ | | | ✓ | |
| BELIA[38] | | | ✓ | | ✓ | ✓ | |
| Wu et al. [217] | | | ✓ | ✓ | | | |
| Trinh et al. [174] | | ✓ | | ✓ | | | |
| Li et al. [223] | | | ✓ | ✓ | ✓ | | |

**7. Some Lessons Learnt**

In this survey, we learned some lessons:

- Until now, whatever work has been done regarding multipath transport policies focuses on the network architecture implemented in the current. However, when we consider the 5G New Radio network, such network architecture will bring about more dynamicity concerning changing path characteristics due to LoS requirements tempted by handover between macro and trivial cells in dense organizations. Thus, in the case of a 5G New Radio network, the problem of implementing a multipath scheduler that will address a higher paths' dynamicity must be considered extensively.
- Several works have been undertaken to promote multipath communication in resource-constrained heterogeneous networks such as IoT, M2M communication [228,229], and vehicular networks.
- More and more multi-homed user devices are equipped with end-to-end multipath communication capability. However, the development of multipath transmission still requires more marketing support to engage industries and users. Currently, only a few companies are making multipath-equipped smartphones. Apple has deployed MPTCP on iPhones; any iOS12 or more applications can use MPTCP as a Layer-4 protocol. LG and Samsung are developing smartphones in South Korea to use cellular and Wi-Fi interfaces to achieve bandwidths of up to 1Gbps [230]. The following major factors/efforts will lead to the optimized development of multipath communication: experience, rigorous testing, fault identification and resolution, government and industry support for the research community, and standardization.
- The limited acceptance of CMT-SCTP by network middleboxes (e.g., port address translation, firewalls, NAT, and so forth) makes MPTCP protocols more acceptable for Internet-based networks. This happens because network middleboxes may force changes in the boundaries of the data stream.
- Most CMT-SCTP and MPTCP multipath protocols address the same problems and include common functional modules such as multipath management and multipath packet schedulers. However, both types of multipath transport protocols struggle to achieve disjointed goals such as fairness, path diversity, pareto-optimality, and receiver buffer impact.
- CMT-SCTP and MPTCP multipath protocols use different control signals for establishing multipath connections. Moreover, they have different CC algorithms that satisfy the requirements for meeting appropriate properties such as load distribution and balancing, delay, bandwidth, the energy consumption of the multiple available paths, QoS metrics, and priority. Moreover, the functional principles of CMT-SCTP and MPTCP protocols differ from each other. Table 6 shows the main differences.

**Table 6.** Comparison of CMT-SCTP and MPTCP protocols.

| Parameter | CMT | MPTCP |
| --- | --- | --- |
| Connection establishment | 4-Way handshaking | 3-Way handshaking |
| CC | Uncoupled | Coupled |
| ACK mechanism | SACK and delay | SACK cumulative ACK, SACK, and delay SACK |
| Compatility of middle boxes | Not compatible | Compatible |
| Performance | High throughput with excessive CPU utilization | Limited throughput |
| Fairness | Limited | Maximum possible |

**8. Open Research Issues**

1.  *Standardization* efforts are required to develop multipath transport protocols rapidly. Only a few basic multipath protocols are standardized [9,44,69,77,79,84,94]. There is still room for much work in this direction.

2.  *Energy consumption* consideration during transmission is also one of the requirements in the resource constraint environment, such as M2M/IoT communication. In some surveys and suggested works [87,121,172–174,217,231–234], energy consumption is taken into consideration during both single and multipath data transmission. Still, there is a dire need for an optimal energy-efficient algorithm to fulfill the future needs of battery-operated multi-homed devices.

3.  *Security in multipath* was discussed in Section 3. However, we could not find any promising work to deal with the security issue. Only [235] presents a relevant research effort on DoS attack handling. Security threats such as handshaking, multiple subflow, flooding, hijacking, and DoS attacks (arising due to multipath transmission) are promising research challenges. Most cyber-attacks usually lack real-time information about various MPTCP attributes. Consequently, considering MPTCP, academia and industry must suggest innovative measurement methods to examine the vulnerability and robustness under cyber-attacks with inadequate network information. Such a measurement technique was introduced recently in [236].

4.  IoT adds more difficulty to multipath communication due to its heterogeneous nature, resource constraint devices, mobility, and dynamic nature. Precisely, implementing MPTCP in IoT systems faces the following technical challenges:

    -   Utilizing different communication methods would incur dissimilar transmission latencies. This might result in a "buffer bloat" at receivers [237]. As a result, this impacts the performance at the Application Layer.

    -   The majority of IoT applications require high QoS demands. Thus, the MPTCP architecture needs to be further improved.

    -   Due to the constant movement of IoT devices (e.g., in a vehicular network), it is hard to preserve a stable network topology in IoT networks. Consequently, it is imperative to design an efficient routing protocol to offer stable communications.

    The above factors are merely considered in the literature. Multipath communication will certainly contribute to the development of IoT significantly. Thus, its impact cannot be ignored.

5.  *Deep-learning and artificial intelligence* (AI) are increasingly becoming key techniques to solve various problems [238]. These techniques can be used to solve the issues which arise in multipath transmission. Several AI-based works have been performed to measure the QoS [239,240] to solve the optimization in single-path transmission. Hence, a better solution could be possible using deep learning, machine learning, or any other artificial intelligence-based methods in solving multipath issues.

6.  Wireless technologies (i.e., Zigbee, Wi-Fi, LoRA, and Bluetooth) and enabled devices are rapidly evolving. Therefore, there is a necessity to discover the influence and evolution of these technologies in multipath transmission [205–207,241,242].

7.  The 5G communication requirements (extremely high bandwidth and ultra-low latency) mandate the multipath transmission capability [243]. Furthermore, some studies incorporated multipath in 5G [244,245]. Therefore, multipath transmission over the 5G network is a key promising research area to innovate solutions required in handling issues of multi-home technologies.

8.  As shown in Table 3, there are several scheduling criteria adopted by multipath schemes. The performance of such schemes hinges on the degree of compatibility of network conditions and applications. It is found that most of the algorithms use a single scheduling criterion and perform better in favorable conditions and poor when conditions change. Therefore, there is a requirement for context-aware scheduling so that the multipath algorithm adopts the best scheduling policy when the context of the network changes [79,246].

9.  The *processing overhead* is increasing more in a multipath transmission than in a single path. Therefore, overhead reduces scalability. We require an optimal solution to minimize the computation power, complexity, and memory use in multipath transmission.

10. Packet reordering, spurious retransmission, and buffering issues are studied in the literature but still pose a great challenge. These are the key factors affecting the performance of multipath communication. Hence, these issues need extra attention in the design of future algorithms.

11. A *cross-layer design (CLD)* methodology is indispensable for multipath transport protocols. This becomes even more prominent and crucial when such protocols are implemented in the wireless network environment (especially in the case of mobile ad-hoc networks where nodes are mobile). A simple idea of CLD methodology is to share dynamic information of crucial factors (such as bit error rate, latency due to path re-computations at the network layer, collision, and network partitions) between non-adjacent layers can meet the demands for high-quality multipath communication [131,214,237,247]. Hence, cross-layer provisioning of multipath transmission can be a promising research area in the future.

Figure 6 summarizes open-research issues identified in this survey.

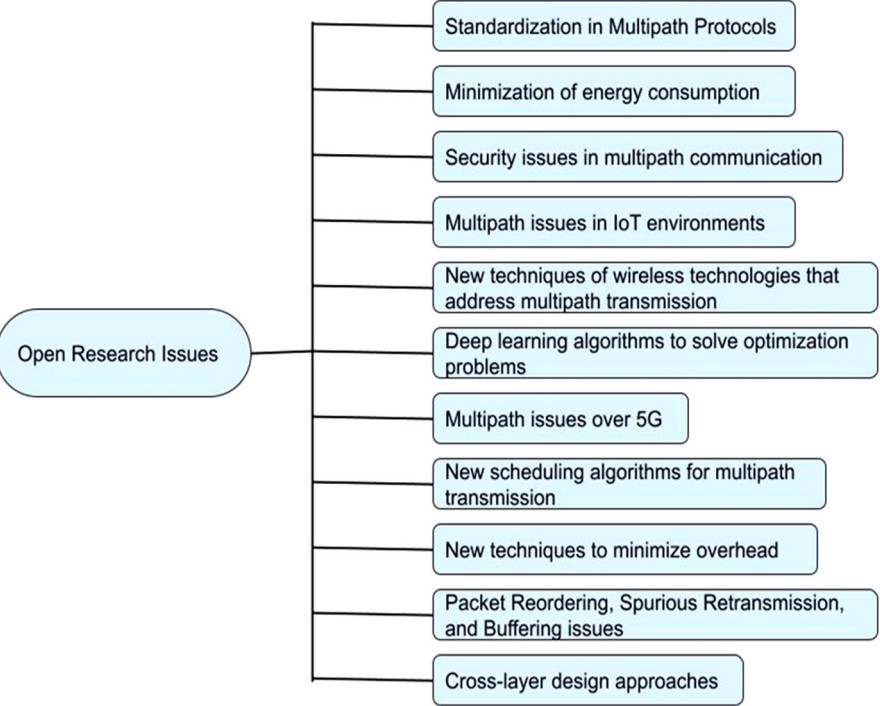

**Figure 6.** Identified open research issues.

## 9. Conclusions

The multipath communication paradigm sustains the growing demand for optimal performance in many networked multimedia applications. Notably, multipath transport protocols can partially meet the requirements of various networked multimedia applications. This survey paper considered how multipath transport protocols (CMT-SCTP and MPTCP variants) are deployed to satisfy various communication scenarios. It also presented key technical challenges/problems in multipath communication. These challenges are multipath scheduling; CWND growth policy; packet loss and retransmission; the RBB problem; excessive network congestion; long RTT; communication channel impairment; heterogeneous communication standards, packet reordering, and stream handling; the HoL blocking problem; the Pareto-optimality issue; and various security issues. It listed all the latest developments of multipath transport protocols along with their functionalities. It reviewed them by considering: (1) how a multipath transport protocol operates (i.e., its functionality); (2) in what type of network; (3) what path characteristics it takes into account; and (4) how it addresses the above design challenges. Furthermore, it identified open research issues in multipath transport protocols. Engineers and protocol developers will find this comprehensive review helpful (a) in the design of multipath transport protocols and (b) in their attempt to increase the performance of a multipath transport protocol for LTE networks.

**Author Contributions:** Conceptualization, P.T.; methodology, G.K., L.P.V. and V.K.S.; analysis and investigation, V.K.S. and D.K.; draft preparation, G.K., L.P.V., V.K.S. and D.K.; supervision, S.S.R. and Y.A. All authors contributed equally. All authors have read and agreed to the published version of the manuscript.

**Funding:** This research received no external funding.

**Institutional Review Board Statement:** Not applicable.

**Informed Consent Statement:** Not applicable.

**Data Availability Statement:** Not applicable.

**Conflicts of Interest:** The authors declare no conflict of interest.

## Abbreviations

The following abbreviations are used in this manuscript:

| | |
|---|---|
| 3GPP | 3rd generation partnership project |
| ACK | Acknowledgement |
| API | Application programming interface |
| ATSSS | Access traffic steering, switching, and splitting (architecture) |
| CC | Congestion control |
| CMT | Concurrent multipath transmission |
| CUMACK | CUMulative ACK |
| CWND | Congestion window |
| CLD | Cross-layer design |
| DCCP | Datagram congestion control protocol |
| DoS | Denial-of-service (attack) |
| FEC | Forward error correction (coding scheme) |
| HoL | Head of line |
| IEEE | Institute of Electrical and Electronics Engineers |
| IETF | Internet engineering task force |
| IoT | Internet of things |
| IP | Internet protocol |
| LoRA | Long range (a spread spectrum modulation technique) |
| LoS | Line-of-sight |
| LTE | Long-term evolution |



| | |
|---|---|
| LIA | "Linked increases" algorithm |
| M2M | Machine to machine |
| MAC | Medium access control (sub-layer) |
| MPTCP | Multi-path transmission control protocol |
| MSD | Mobile smart device |
| NAT | Network address translator |
| NC | Network coding |
| OSI-RM | Open systems interconnection—reference model |
| QoE | Quality of experience |
| QoS | Quality of service |
| RBB | Receiver buffer blocking |
| RFC | Request for comments |
| RL | Reinforcement learning |
| RSN | Route sequence number |
| RTT | Round-trip-time |
| SACK | Selective acknowledgments |
| SCTP | Stream control transmission protocol |
| ssthresh | Slow start threshold |
| TCP | Transmission control protocol |
| TSN | Transmission sequence number |
| VANET | Vehicular ad hoc network |

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
