# Peer review of "CMT-SCTP and MPTCP Multipath Transport Protocols: A Comprehensive Review"

_electronics, doi:10.3390/electronics11152384_

Round 1
Reviewer 1 Report
1. Regarding the proposed hasting algorithm, lacking descriptions on contribution manner and its functionality obligated extending distinguish.
2. The image preprocessing phase consists of insufficient elaboration
3. The general definitions of index terms should be lessened to strengthen paper to meet technical sound.
4. author should provide further descriptions of the paper contributions and detailed reason why the proposed approach enhances existed works.
5. Acronyms, the authors haven’t written the full meaning of the word. For example
word (CWND) at line 75
6. Acronyms, the authors haven’t written the full meaning of the word. For example,
word (ATSSS) at line 133
7. Acronyms, the authors haven’t written the full meaning of the word. For example,
word (QUIC) at line 138
8. Acronyms, the authors haven’t written the full meaning of the word. For example,
word (FMTCP) at line 278
Reviewer 2 Report
The authors present the article entitled “CMT-SCTP and MPTCP Multipath Transport Protocols: A Comprehensive Review”
This survey paper focuses on connection-oriented multipath protocols located at the Transport Layer of the Open Systems Interconnection-Reference Model OSI-RM.
The article presents the following concerns:
Abstract: Please improve this section by mentioning the main conclusions or interpretations.
Vectorize the figures in order to see the details.
447-469: The author is referenced many times. I suggest improving the idea.
Line 906-911: I recommend improving the sentence. In its current form, it is hard to follow because it first mentions [190] then [191] and finally back to [190] in different sentences. Check if it is repeated in other paragraphs.
The description of tables 1 and 2 in the main text is poor, please improve it.
Improve the discussion of Figure 6.
line 309: Check the guide for authors for grouped references.
Introduction section is very extensive. I suggest to present subsection 1.1 as another section
I recommend giving an introduction between section 2 and 2.1., 4 and 4.1, and 5 and 5.1
The subtitle 5.2.1 is not equal to other subtitles. I recommend you do it in the same form in all text
Apostrophes must be avoided
The text must be written in the 3rd person or passive voice.
Line 1011 could be justified with the next examples regarding artificial intelligence usage:
Implementation of ANN-Based Auto-Adjustable for a Pneumatic Servo System Embedded on FPGA
Fuzzy Logic and Genetic-Based Algorithm for a Servo Control System
Spatial Models and Neural Network for Identifying Sustainable Transportation Projects with Study Case in Querétaro, an Intermediate Mexican City
The following misspelling should be checked:
line 159: “the exploitation of…” should be rewritten as “exploiting…”
line 227: It appears that “up” may be unnecessary in this sentence. consider removing it.
line 230: What does ssthresh? It’s the correct form?
line 286: The subordinate phrase “To alleviate this problem” does not appear to be modifying the subject “different techniques”. Rewite the sentence to avoid a dangling modifier.
line 438: The phrase “is entirely dependent” should be rewritten as “depends entirely”. Consider changing.
line 470: You need to add the article "an" to “unordered”
line 528: The phrase “delivery of data” must be changed by “data delivery”
line 666: “Although through…” It seems that you are missing a comma. Consider adding “Although, through…”
Round 2
Reviewer 1 Report
The paper is well revised and can be accepted.